# Sex-Specific Cut-Offs of Single Point Insulin Sensitivity Estimator (SPISE) in Predicting Metabolic Syndrome in the Arab Adolescents

**DOI:** 10.3390/diagnostics13020324

**Published:** 2023-01-16

**Authors:** Kaiser Wani, Malak N. K. Khattak, Gamal M. Saadawy, Omar S. Al-Attas, Majed S. Alokail, Nasser M. Al-Daghri

**Affiliations:** Department of Biochemistry, College of Science, King Saud University, Riyadh 11451, Saudi Arabia

**Keywords:** SPISE, metabolic syndrome, insulin sensitivity, insulin resistance, sex-specific, resistin, leptin, adiponectin, vitamin D

## Abstract

The Single Point Insulin Sensitivity Estimator (SPISE) is a novel surrogate marker for insulin sensitivity and was found comparable to the gold standard clamp test as well as for predicting the Metabolic Syndrome (MetS) in several populations. The present study aimed to assess for the first time, the validity of SPISE in predicting MetS among Arab adolescents. In this cross-sectional study, 951 Saudi adolescents aged 10–17 years were randomly recruited from different schools across Riyadh, Saudi Arabia. Anthropometrics were measured and fasting blood samples were collected for the assessment of glucose, lipid profile, adipokines, C-reactive protein and 25 hydroxyvitamin (OH) D. MetS was defined using the National Cholesterol Education Program’s (NCEP) criteria with age-specific thresholds for adolescents. The SPISE as well as insulin resistance (HOMA-IR) indices were calculated. The over-all prevalence of MetS was 8.6% (82 out of 951). SPISE index was significantly lower in MetS than non-MetS participants in both sexes (5.5 ± 2.5 vs. 9.4 ± 3.2, *p* < 0.001 in boys and 4.4 ± 1.4 vs. 8.6 ± 3.2, *p* < 0.001 in girls). The SPISE index showed a significant inverse correlation with resistin, leptin, and C-reactive protein, and a significant positive correlation with adiponectin and 25(OH) D. Areas under the curve (AUC) revealed fair and good accuracy for predicting MetS 84.1% and 90.3% in boys and girls, respectively. The sex-specific cut-off proposed was SPISE index ≤6.1 (sensitivity 72.2% and specificity 83.9%) for boys and ≤6.46 (sensitivity 96.3% and specificity 73.4%), for girls. This study suggests that the SPISE index is a simple and promising diagnostic marker of insulin sensitivity and MetS in Arab adolescents.

## 1. Introduction

Metabolic syndrome (MetS), a combination of several cardiometabolic risk factors such as obesity, hypertension, hyperglycemia, and dyslipidemia, is highly prevalent in Saudi Arabia, with a recent study involving more than 12,000 adults estimating the prevalence to be at 40% [1]. Multiple prospective studies have reported that the presence of MetS is associated with major cardiovascular events and all-cause mortality, and the presence of MetS alone can predict 25% of all new-onset cardiovascular diseases (CVD) [2,3]. While MetS is more common in middle-aged and older adults, recent observations indicate that children and adolescents are increasingly more likely to have MetS because of unhealthy lifestyles such as being sedentary and frequent intake of calorie-dense fast foods [4,5]. This may explain the increase in the prevalence in childhood obesity and MetS, especially in developing countries such as Saudi Arabia [6,7,8]. A recent meta-analysis revealed that early intervention, targeting children and adolescents in the school-setting, may prove beneficial in reducing the prevalence of obesity and MetS [9]. This early interventional approach may prove beneficial since genetically the Arab ethnicity is highly susceptible to DM, a consequence of MetS, and is heritable [10].

Insulin resistance, defined as decreased responsiveness to the metabolic actions of insulin, has been widely blamed for MetS and its individual risk factors, driving the European Group for the study of Insulin Resistance (EGIR) to propose the term insulin resistance syndrome for MetS [11]. The role of insulin in glucose hemostasis in several tissues has been observed to play a complex role in cellular mechanisms including endoplasmic reticulum stress, lipotoxicity, reactive oxygen species and inflammatory signaling, among others [12]. Another important factor is the amount of visceral adipose tissue, which acts as a complex endocrine organ. It secretes a number of adipokines, some of which are proinflammatory and atherogenic such as leptin and resistin, while some others such as adiponectin have anti-inflammatory effects [13,14]. Through their effects on the endocrine system, these factors help control appetite, thermogenesis, glucose metabolism, and lipid metabolism and thus contribute to systemic energy metabolism [14]. The imbalance in these adipose-tissue derived factors can disrupt health systemic physiology, with type 2 diabetes and cardiovascular disease among the consequences. Similarly, there are reports which have linked vitamin D deficiency with impaired fasting glucose (IFG) characterized by IR and β-cell dysfunction [15].

Regardless of whether it is the sole causative agent or a result of MetS and its individual cardiometabolic risk factors, determining insulin sensitivity is important, and many methods have been proposed, ranging from complex techniques to simple indices [16,17]. The euglycemic hyperinsulinemic clamp test has been a gold standard for over four decades and is a direct method for estimation of IR; however, it is labor-intensive, impractical and uncomfortable for the patient [18]. Simpler indices utilizing circulating fasting glucose, fasting insulin, and triglyceride levels, such as the homeostasis model assessment for insulin resistance (HOMA-IR), the quantitative insulin sensitivity check index (QUICKI), the McAuley index and the Matsuda index were proposed and used over the years despite some caveats such as huge variability in insulin assays [19].

A lipid-based index called the triglyceride/HDL-cholesterol (TG/HDL) ratio is a simpler surrogate marker for insulin sensitivity and one that can be available in routine clinical settings; it was introduced but discovered to have large variability in the proposed cut-offs [20,21]. Paulmichl et al. [22] refined this ratio using a computer-assisted mathematical modeling to an index termed as Single Point Insulin Sensitivity Estimator (SPISE), which was found to have better sensitivity and specificity than traditional indices when compared with the M-value of <4.7 mg/kg/min, the euglycemic hyperinsulinemic clamp test cut-off for higher IR. The SPISE index was developed from a cohort of 1300 adults recruited from 14 European countries [22] and later validated in other populations such as Korean adolescents [23], Hispanic adolescents [24], and Indian adults [25]. As a novel surrogate marker of insulin sensitivity, further validation in other populations is needed as well. We hypothesize that it may also serve as a surrogate marker of insulin sensitivity in this population but with a different cut-off than those reported in other populations. The present study aims to validate SPISE and assess its optimal cut-off in Arab adolescents with MetS.

## 2. Materials and Methods

### 2.1. Study Participants and Recruitment

In this cross-sectional study, Saudi adolescents (age range 10–17 years) were randomly recruited from 60 schools across Riyadh, Saudi Arabia, from September 2019 to March 2021. Non-Saudis and those outside the age range were excluded as well as anyone with debilitating acute and chronic conditions (e.g., Addison’s disease, Cystic fibrosis, Graves’ disease, irritable bowel syndrome, chronic kidney disease, etc.). Those with known genetic syndromes including syndromic obesity were excluded. The study was a collaborative project by the Chair for Biomarkers of Chronic Diseases (CBCD) at King Saud University (KSU) and the Saudi Charitable Association for Diabetes (SCAD) to educate school-aged children about unhealthy dietary habits, sedentary behaviours, physical inactivity, and the rising prevalence of obesity and metabolic syndrome in Saudi adolescents [6,26,27]. The institutional review board (IRB) of the College of Medicine at King Saud University granted ethical approval (No. E-19-4239) and the study was conducted in accordance with ethical guidelines for research involving children. All participants engaged in the study provided written informed parental consent and assent before inclusion.

### 2.2. Clinical and Biochemical Evaluations

All recruited study participants underwent clinical examination, anthropometric assessment and fasting blood sample collection. A standard questionnaire with demographic information (age, sex, medical history and school) was filled out by each participant. Anthropometric measurements included weight (kg), height (cm), waist and hip circumferences (cm) and systolic and diastolic blood pressures (mmHg) were recorded by trained research nurses. Body mass index (BMI) was calculated (kg/m^2^) and recorded. Fasting blood samples were collected from each participant, processed, aliquoted, and transported to the CBCD laboratory for biochemical evaluations. A routine biochemistry analyzer (Konelab 20XT, Thermo Scientific, Vantaa, Finland) was used to measure the circulating levels of glucose, total cholesterol, HDL-cholesterol and triglycerides using commercially available bioassay kits (reference# 981379, 981812, 981823, and 981301, respectively). Another automated quantitative analyzer, COBAS e-411 (Roche Diagnostics, Indianapolis, IN, USA), was used to quantify serum 25(OH) D levels. The assay had a range of 7.5–250 nmol/L and an inter- and intra-assay CV of <5.5% and <7.0%, respectively. Luminex multiplex (Luminexcorp, Austin, TX, USA), was used in analyzing a selected set of adipokines. The bioassay kit# HADK1MAG-61 K was used to measure adiponectin, resistin, and APAI-1 with inter-assay and intra-assay variations of <10% and <15%, respectively. Another bioassay kit, HBNMAG-51 K, based on the same multiplex technology, was used to measure circulating levels of insulin, leptin and tumor necrosis factor alpha (TNF-α) [28]. Circulating C-reactive protein (CRP) levels were determined using commercially available ELISA assay kits (My BioSource, San Diego, CA, USA; catalog numbers: MBS2505217, with intra- and inter-assay CVs of 3.95% and 6.07%, respectively).

### 2.3. MetS Components and SPISE Index Determination

Screening for MetS was conducted using the National Cholesterol Education Program’s (NCEP) criteria [≥3 out of 5 MetS components, namely, elevated waist circumference, elevated blood pressure, elevated fasting glucose, elevated triglycerides, and low HDL-cholesterol was categorized as having MetS and those with ≤2 MetS components were categorized as non-MetS]. Different thresholds for pediatric MetS were taken from the criteria given by Cook et al. [29] as below:Elevated waist circumference: age-specific waist circumference of ≥90th percentileElevated blood pressure: age-specific systolic or diastolic blood pressure of ≥90th percentileElevated fasting glucose: fasting glucose level of ≥6.1 mmol/LElevated triglycerides: circulating triglyceride levels of ≥1.24 mmol/L for age 10–15 years and ≥1.7 mmol/L for age ≥16 yearsLow HDL-cholesterol: circulating HDL-cholesterol level of ≤1.03 mmol/L

Insulin sensitivity and resistance were estimated by calculating validated indirect indexes. Insulin sensitivity was assessed by the SPISE index, calculated as SPISE index = (600 × (HDL − Cholesterol in mg/dL^0.185^))/(Triglyceride in mg/dL^0.2^ × BMI in kg/squarem^1.338^) [22]. The “Insulin resistance index” used was “Homeostasis Model Assessment for Insulin Resistance” (HOMA-IR) calculated as: ((HOMA-IR: fasting insulin in µU/L × fasting glucose in nmol/L)/22.5) [30].

### 2.4. Data Analysis

The data, including clinical and biochemical groupings into individual components of MetS and the SPISE index values, was compiled and analyzed using the Statistical Package for Social Sciences (SPSS) version 23.0. For categorical, normal continuous, and non-normal continuous variables, data was presented as frequency (%), mean standard deviation, or median (25th percentile–75th percentile), and differences in these variables between boys and girls were tested using the chi-square test, parametric and non-parametric independent student *t*-tests, respectively. The subjects were then divided between MetS and non-MetS categories, and the differences in variables apart from those used to categorize the subjects were checked between the groups for the overall data and also within sexes. The differences in the SPISE index were also checked between sexes and between MetS and non-MetS groups within sexes. A bivariate correlation analysis was run to check the association between the SPISE index and measured clinical parameters. A ROC curve analysis was also performed using the SPISE index to predict MetS from non-MetS in our subjects, and the sex-specific cut-offs in the SPISE index were also reported. The subjects were then divided based on these SPISE index cut-offs, and the prevalence of MetS components in these groups was analyzed and reported.

## 3. Results

### 3.1. Clinical Characteristics of the Subjects

The clinical characteristics of the study subjects at the time of recruitment are presented in Table 1. The average age of boys and girls recruited was comparable (13.9 ± 2.2 years in boys vs. 13.7 ± 2.3 years in girls, *p* = 0.14). Girls had a significantly higher BMI and lower waist hip ratio (WHR) than boys. The adipokines adiponectin, resistin and leptin levels were significantly higher in girls than boys, while the average 25(OH) D levels in boys were higher in boys than girls. The adiponectin/leptin ratio was also significantly higher in boys than girls (*p* < 0.001). The SPISE index in boys was significantly higher than girls (8.95 ± 3.3 vs. 8.37 ± 3.4, *p* = 0.01).

### 3.2. Characteristics of the Study Subjects Divided into MetS/Non-MetS Groups

The clinical characteristics of participants according to MetS status are shown in Table 2. The overall prevalence of MetS in the study participants was 8.6% (10.7% in boys and 6.3% in girls). Expectedly, those with MetS had a significantly higher BMI, were older, had higher HOMA-IR values, and had significantly higher average circulating levels of C-reactive protein (CRP), leptin and insulin. The adiponectin/leptin ratio was significantly lower in the MetS group than the non-MetS group. The SPISE index was significantly lower in the MetS group than the non-MetS group (5.5 ± 2.5 vs. 9.4 ± 3.2, *p* < 0.001).

The average SPISE values in subjects with 0, 1, 2 and >2 components of MetS have been plotted (Figure 1). Participants with none to three MetS components showed a decreasing trend in SPISE values, independent of sex.

### 3.3. Associations of SPISE Index with Other Measured Variables

Bivariate associations of SPISE index with other measured parameters were presented in Table 3. A significant inverse correlation was seen between the SPISE index and age, resistin, leptin, CRP, insulin, and HOMA-IR, and a significant positive correlation with adiponectin, 25(OH) D and adiponectin/leptin ratio. A similar significant correlation was observed after stratification according to sex.

The significant bivariate correlations between SPISE index and select clinical variables are shown in Figure 2.

### 3.4. Receiver Operating Characteristic (ROC) Curve of SPISE Index in Predicting MetS/Non MetS

The SPISE index showed high specificity and sensitivity in predicting the MetS in both sexes (Figure 3). The area under the curve (AUC) for the SPISE index in predicting MetS in boys and girls was 84.1% and 90.3%, respectively, and the Youden cut-off for best sensitivity and specificity in this analysis was the SPISE index ≤6.14 and 6.46, for boys and girls, respectively. The coordinates of the analysis for all participants have been provided in the Appendix A. Coordinates for males and females were also included as Appendix A, respectively.

### 3.5. MetS Components According to Cut-Offs in SPISE Index

Participants were stratified according to obtained SPISE index cut-offs and logistic regression was used to compare proportions in these groups (Table 4). Among the components, low HDL-cholesterol was most prevalent (in 53.7% of boys and 47.3% of girls) followed by hypertriglyceridemia (in 22.5% of boys and 21% of girls). The prevalence of all MetS components, with the exception of hyperglycemia in boys, was significantly higher in groups with a SPISE index ≤ cut-off than the group with a SPISE index > cut-off, irrespective of sex, with an age-standardized odds ratio (95% CI) ranging from as low as 1.81 (1.2, 2.8) in the case of low HDL-cholesterol in girls to as high as 29.72 (14.1, 62.7) in the case of central obesity in boys.

## 4. Discussion

The main findings in this study suggest that the SPISE index is a clinically useful indicator of insulin sensitivity in Arab adolescents with MetS. Sex-specific cut-offs obtained had good to excellent sensitivity and specificity for predicting pediatric MetS in this population.

Insulin resistance predisposes individuals to hyperglycemia, hypertension, and dyslipidemia, which are the hallmarks of MetS [31,32]. Innovative approaches to prevent the onset of MetS are needed to counteract the rising prevalence, particularly in populations where MetS and obesity are becoming more common even in children and adolescents [27,33]. The euglycemic hyperinsulinemic clamp (EHC) developed by DeFronzo et al. in 1979, is the gold standard method for measuring IR [34]. However, this glucose clamp approach has limited clinical use since it is more labor-intensive, complicated and costly. Over time, a number of clinically useful surrogate measures of IR have been developed and validated using the gold standard method [35,36,37]. Lipid indices such as the TG/HDL ratio [38], later modified by Paulmichl et al. [22] and known as the SPISE index, were later developed and showed better sensitivity and specificity when validated with the gold standard for measuring IR.

In this study, the SPISE index in Arab adolescents with MetS compared to those without MetS had AUC values of 84.1% and 90.3%, respectively, in boys and girls (*p*-values < 0.001 in both sexes). The results are in line with the results of previous studies where this surrogate marker was calculated and validated for IR with AUC ranging 66.5–83% [22,23,24,25]. Among the early signs of IR are abnormalities in lipid and lipoprotein metabolism, which may account for the high AUC value for this lipid-based insulin sensitivity index in identifying IR and hence MetS [39]. Increased availability of free fatty acids as well as hyperglycemia are associated with a rise in very-low-density lipoprotein (VLDL) cholesterol levels, apart from lower levels and smaller particle sizes of HDL-cholesterol in IR and MetS states, as shown in NMR spectroscopic investigations [40,41]. IR and therefore MetS are often accompanied with the key physiological characteristic of dyslipidemia, an increase in the production of triglycerides in very low density lipoproteins (VLDL), and therapy of VLDL has been shown to reduce the risk of cardiovascular disease [42]. Insulin regulates lipoprotein lipase (LPL), an enzyme that destroys triglyceride levels in the blood, and LPL levels do decrease somewhat in IR and MetS states [43,44].

The obtained Youden cutoffs of the SPISE index in Arab boys and girls vary from the previously reported cut-offs conducted in different populations [22,23,24,25]. Aside from participants’ age and other methodological differences, the variances may be attributed differences in ethnicities [45,46]. Although not as extreme as earlier surrogate markers, differences in the reported cut-offs of the novel SPISE index necessitate the need to validate this index in different ethnicities, as was the case with its predecessor, the TG/HDL ratio [47].

The correlation analysis of the SPISE index with other measured parameters conducted in this study revealed significant inverse correlations with adipose tissue-derived hormones such as resistin and leptin and inflammatory markers such as CRP. Resistin, for instance, may interfere with insulin signaling by promoting the expression of phosphatase and tensin homologue (PTEN) [48]. Leptin, on the other hand, well known for its role in body weight regulation, has also been reported to be independently involved in the regulation of glucose homeostasis and hence IR [49]. Consequently, the recently proposed “adiposity-based chronic disease (ABCD)”, takes into account not only the total amount of obesity but also its distribution and function [50]. Apart from the mentioned associations, some other probable mechanisms through which adipose-tissue related factors influence IR may include their involvement in glucose metabolism and lipolytic activity [51,52].

Similarly, the link between endothelial inflammatory markers such as CRP and IR has been extensively studied and published, lending credibility to the observed relationships between CRP and the SPISE index in our study [53,54]. The bivariate correlation analysis conducted in this study also revealed a significant positive association of the SPISE index with circulation levels of adiponectin and vitamin D. Decades of research on adiponectin established its role in glucose uptake and fat storage, leading to increased insulin sensitivity and inflammation [55]. Similarly, several interventions of vitamin D supplementation suggest improvement in insulin sensitivity post supplementation [56]. Additionally, adiponectin/leptin ratio, a marker of adipose tissue inflammation, and SPISE index had a strong positive link, which is consistent with past research suggesting that lower adiponectin/leptin levels are predictive of cardiometabolic risk [57].

The SPISE index decreased significantly with increased MetS components in both sexes. Additionally, the prevalence of individual MetS components was significantly higher in individuals with a SPISE index ≤ cut-off compared to ones with a greater than cut-off, and this was observed independent of sex. The logistic regression also suggested that the odds of having all components of MetS in subjects with a SPISE index ≤ cut-off were not only higher within sexes but also markedly higher for central obesity between sexes (boys, OR of 29.72, *p* < 0.001 vs. girls, OR of 7.25, *p* < 0.001). The differences observed here complement the known sex differences in genetic makeup and endogenous sex hormones and their roles in obesity, diabetes, CVD, and HOMA-IR [58,59]. This further reinforces the significance of the SPISE index as a valuable marker of early morphological and physiological vascular disturbances, a good predictor, and a possible low-cost, routinely available simple index of MetS.

The authors acknowledge several limitations. First, the cross-sectional design of the study prevents causality. Participants were within the age range of 10–17 years, which makes the results not fully representative of Saudi children and other studies may be needed to determine the cut-offs in the mentioned sub-cohorts. Furthermore, evaluation of the SPISE index’s reliability as an IR marker using the euglycemic hyperinsulinemic test was not conducted. Lastly, the data on pubertal developmental stages, as proposed by Marshall and Tanner [60], were not collected in this study and may have had an impact on the SPISE index values, which should be investigated in future studies.

## 5. Conclusions

In conclusion, this study suggests that the SPISE index is a clinically useful indicator of insulin sensitivity for predicting pediatric MetS in Arab populations. While this novel index is a promising diagnostic marker of insulin sensitivity and MetS in Arab adolescents, obtained cut-offs must be validated with the gold standard clamp test.

## Figures and Tables

**Figure 1 diagnostics-13-00324-f001:**
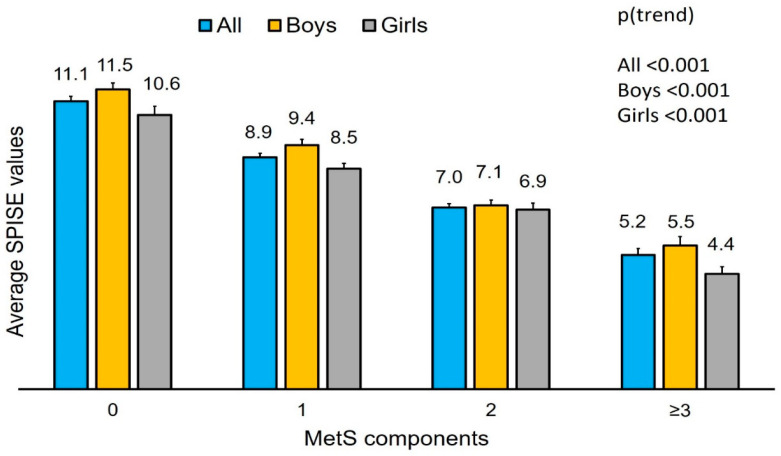
Average SPISE values for subjects with different MetS components.

**Figure 2 diagnostics-13-00324-f002:**
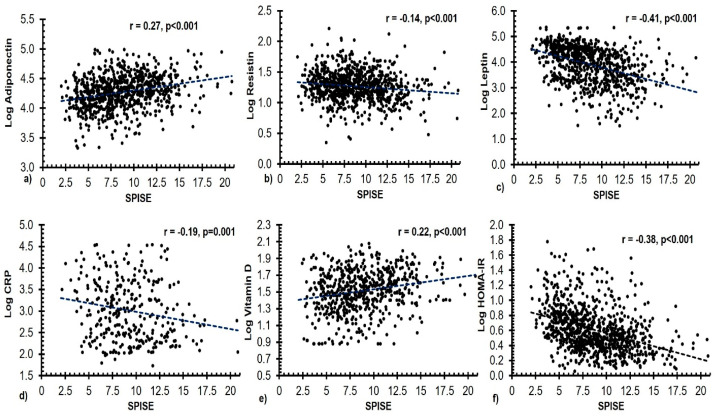
Scatter plots depicting significant bivariate correlations between the SPISE index and the circulating levels of (**a**) adiponectin (ng/mL), (**b**) resistin (pg/mL), (**c**) leptin (pg/mL), (**d**) CRP (ng/mL), (**e**) 25(OH) D (nmol/L), and (**f**) HOMA-IR in all subjects.

**Figure 3 diagnostics-13-00324-f003:**
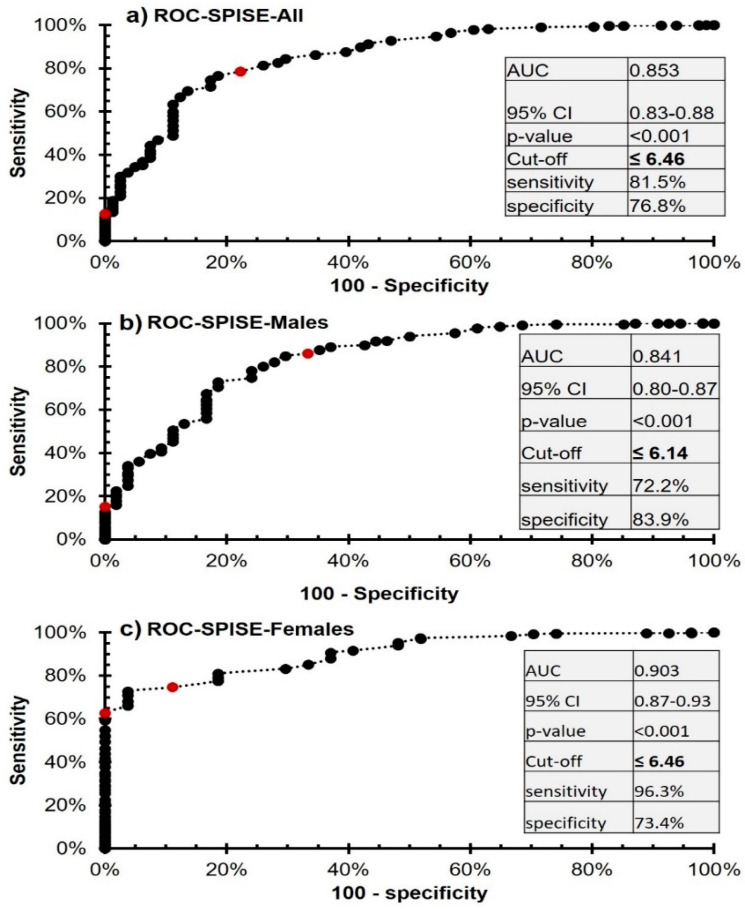
ROC analysis of the SPISE index for the prediction of MetS from non-MetS in Saudi adolescents.

**Table 1 diagnostics-13-00324-t001:** Clinical and Anthropometric characteristics of study participants at recruitment.

Parameters	All Subjects (951)	Boys (503)	Girls (448)	*p*-Value
Anthropometric Characteristics				
Age (years)	13.8 ± 2.3	13.9 ± 2.2	13.7 ± 2.3	0.14
BMI (kg/m^2^)	22.1 ± 6.3	21.4 ± 6.2	22.8 ± 6.3	0.001
Waist (cm)	75 ± 16	76 ± 17	74 ± 15	0.06
Hip (cm)	87 ± 17	86 ± 18	89 ± 16	0.001
WHR	0.88 ± 0.3	0.92 ± 0.4	0.84 ± 0.1	<0.001
Systolic BP	107 ± 10	107 ± 10	106 ± 10	0.35
Diastolic BP	69 ± 7	69 ± 8	69 ± 7	0.96
Lipid Profile				
Total Cholesterol (mmol/L)	4.2 ± 0.8	4.2 ± 0.8	4.2 ± 0.8	0.35
HDL-Cholesterol (mmol/L)	1.02 ± 0.4	0.99 ± 0.3	1.1 ± 0.3	0.007
Triglycerides (mmol/L)	0.9(0.7–1.3)	0.96(0.7–1.3)	0.94(0.7–1.3)	0.69
Adipocytokines and Inflammatory Markers				
Adiponectin (µg/mL)	18.9(12.5–28.5)	17.8(11–27.7)	20.1(12.8–29.7)	0.001
Resistin (ng/mL)	18.6(13.5–25.6)	17.6(12.1–23.8)	19.9(14.8–27.9)	<0.001
Leptin (ng/mL)	11.4(2.5–31.4)	6.3(1.8–22.2)	19.5(5.4–38.1)	<0.001
Adiponectin/Leptin	1.82(0.5–8.9)	2.99(0.7–13.3)	1.17(0.4–4.2)	<0.001
TNF-Alpha (pg/mL)	8.4(5.7–12.1)	9.1(6.0–13.2)	7.7(5.3–10.6)	<0.001
CRP (µg/mL)	0.99(0.3–3.9)	0.99(0.3–3.7)	1.01(0.3–4.1)	0.89
APAI-1 (ng/mL)	24.2(14.5–34.5)	23.7(12.9–34.3)	24.3(15.3–34.7)	0.35
25(OH) D (nmmol/L)	33.4(22.7–48.0)	40.9(29.5–57.5)	26.4(17.6–37.3)	<0.001
Glycemic Profile				
Glucose (mmol/L)	5.1 ± 1.1	5.1 ± 1.0	5.1 ± 1.2	0.79
Insulin (miU/mL)	11.8(6.9–21.3)	10.9(6.3–21.3)	12.5(7.7–21.4)	0.027
HOMA-IR	2.6(1.5–4.9)	2.4(1.3–5.0)	2.8(1.7–4.9)	0.052
SPISE	8.67 ± 3.3	8.95 ± 3.3	8.37 ± 3.4	0.010

Note: For normal and non-normal variables, data are presented as Mean ± SD and median (1st quartile–3rd quartile). The difference in these variables between boys and girls was tested by independent student *t*-tests, and a *p*-value < 0.05 was considered significant.

**Table 2 diagnostics-13-00324-t002:** Clinical Characteristics of Participants according to MetS status.

Parameters	All (951)			Boys (503)			Girls (448)		
	Non-MetS 869	MetS 82	*p*-Value	Non-MetS 449	MetS 54	*p*-Value	Non-MetS 420	MetS 28	*p*-Value
Age (years)	13.7 ± 2.3	14.9 ± 1.9	<0.001	13.8 ± 2.2	14.7 ± 2.1	0.003	13.6 ± 2.3	15.2 ± 1.8	<0.001
BMI (kg/m^2^)	21.4 ± 5.7	28.9 ± 7.8	<0.001	20.7 ± 5.6	27.6 ± 7.9	<0.001	22.2 ± 5.7	31.9 ± 7.0	<0.001
Adiponectin (µg/mL)	19.6 (13.1–29.5)	13.4 (9.9–17.5)	<0.001	18.5 (11–28.4)	13.1 (9.5–18.5)	<0.001	20.3 (14–29.0)	14.8 (10.2–18.6)	0.002
Resistin (ng/mL)	18.6 (13.4–25.6)	18.9 (13.9–24.7)	0.79	17.5 (12.1–24.0)	18.4 (12.9–22.9)	0.96	19.8 (14.7–27.7)	22.8 (15.9–32.6)	0.32
Leptin (ng/mL)	10.6 (2.4–29.9)	22.7 (6.9–48.9)	<0.001	5.2 (1.7–20.2)	15.7 (5.6–42.2)	<0.001	18.4 (5.3–36.6)	38.8 (20.5–69.1)	0.004
Adiponectin/Leptin	1.96 (0.6–9.7)	0.55 (0.3–2.6)	<0.001	3.95 (0.8–14.4)	0.74 (0.3–2.9)	<0.001	1.28 (0.5–4.5)	0.31 (0.2–0.6)	0.001
TNF-Alpha (pg/mL)	8.3 (5.6–11.7)	10.4 (6.1–15.1)	0.005	8.9 (6.0–13.0)	10.9 (6.3–14.7)	0.09	7.6 (5.3–10.2)	9.8 (5.4–15.2)	0.06
CRP (µg/mL)	0.97 (0.3–3.8)	3.2 (0.5–10.2)	<0.001	0.97 (0.3–3.7)	2.03 (0.3–14.7)	<0.001	0.98 (0.3–3.9)	5.2 (3.2–10.2)	<0.001
APAI-1 (ng/mL)	24.2 (14.6–34.5)	24.9 (11.1–38.4)	0.94	23.8 (14–34.4)	22.4 (96.4–32.5)	0.33	24.2 (15.3–34.4)	29.3 (19.4–42.5)	0.10
Vitamin D (nmmol/L)	33.5 (22.7–48.0)	33.1 (20.5–57.9)	0.94	40.8 (29.5–55.7)	40.9 (30.6–62.3)	0.35	27.2 (18.3–38.0)	19.0 (8.6–25.5)	0.006
Insulin (miU/mL)	11.1 (6.6–19.7)	26.4 (14.1–46.5)	<0.001	10.1 (5.8–18.5)	28.5 (12.9–44.4)	<0.001	12.1 (7.6–20.2)	23.9 (14.5–60.7)	<0.001
HOMA-IR	2.5 (1.4–4.5)	6.1 (3.2–13.5)	<0.001	2.2 (1.3–4.3)	5.9 (3.2–13.5)	<0.001	2.7 (1.6–4.6)	6.2 (2.9–13.7)	<0.001
SPISE	9.04 ± 3.2	5.17 ± 2.3	<0.001	9.41 ± 3.2	5.53 ± 2.5	<0.001	8.64 ± 3.2	4.44 ± 1.4	<0.001

Note: Data presented as Mean ± SD and median (1st quartile–3rd quartile) for normal and non-normal variables, respectively. The difference in these variables between MetS and non-MetS groups were tested by independent student t-tests and *p*-value < 0.05 were considered significant.

**Table 3 diagnostics-13-00324-t003:** Bivariate correlation analyses between SPISE index and measured clinical parameters.

Parameters	All Subjects		Boys		Girls	
	Correlation Coefficient	*p*	Correlation Coefficient	*p*	Correlation Coefficient	*p*
Adiponectin	0.27	<0.001	0.22	<0.001	0.36	<0.001
Resistin	−0.14	<0.001	−0.17	<0.001	−0.08	0.098
Leptin	−0.41	<0.001	−0.39	<0.001	−0.43	<0.001
Adiponectin/Leptin	0.49	<0.001	0.45	<0.001	0.51	<0.001
TNF-Alpha	0.06	0.08	−0.03	0.60	0.14	0.005
CRP	−0.19	0.001	−0.24	0.001	−0.13	0.12
APAI-1	−0.04	0.25	−0.02	0.63	−0.05	0.31
Vitamin D	0.22	<0.001	0.13	0.027	0.27	<0.001
Insulin	−0.40	<0.001	−0.42	<0.001	−0.37	<0.001
HOMA-IR	−0.38	<0.001	−0.41	<0.001	−0.35	<0.001

Note: The Pearson correlation coefficient and associated *p*-values are used to represent the data. The non-gaussian variables were log-transformed for normality before the analyses. *p*-value <0.05 were considered significant.

**Table 4 diagnostics-13-00324-t004:** Prevalence of MetS components in groups formed using cutoff in SPISE index.

**Boys (503)**							
**MetS Components**	**Present**	**SPISE ≤ 6.14 (103)**	**SPISE > 6.14 (349)**	**OR (95% CI)**	***p*-Value**	**OR (95% CI) ***	***p*-Value ***
Elevated waist circumference	62 (12.3)	52 (50.5)	10 (2.9)	34.56 (16.5, 72.3)	<0.001	29.72 (14.1, 62.7)	<0.001
Elevated blood pressure	61 (12.1)	27 (26.2)	34 (9.7)	3.29 (1.9, 5.8)	<0.001	2.93 (1.6, 5.2)	<0.001
Elevated fasting glucose	55 (10.9)	20 (19.4)	35 (10.0)	2.16 (1.2, 3.9)	0.012	1.81 (1.0, 3.4)	0.07
Elevated triglycerides	113 (22.5)	45 (43.7)	68 (19.5)	3.21 (2.1, 5.1)	<0.001	3.51 (2.2, 5.7)	<0.001
Low HDL-Cholesterol	270 (53.7)	91 (88.3)	179 (51.3)	2.51 (1.5, 3.6)	<0.001	2.29 (1.2, 2.6)	<0.001
MetS	54 (10.7)	39 (37.9)	15 (4.3)	13.56 (7.1, 26.1)	<0.001	12.37 (6.3, 24.1)	<0.001
**Girls (448)**							
**MetS Components**	**Present**	**SPISE ≤ 6.46 (128)**	**SPISE > 6.46 (282)**	**OR (95% CI)**	***p*-Value**	**OR (95% CI) ***	***p*-Value ***
Elevated waist circumference	59 (13.2)	44 (34.4)	15 (5.3)	9.32 (4.9, 17.6)	<0.001	7.25 (3.7, 14.2)	<0.001
Elevated blood pressure	61 (13.6)	29 (28.2)	32 (9.2)	2.29 (1.3, 3.9)	0.004	2.31 (1.3, 4.1)	0.004
Elevated fasting glucose	43 (9.6)	24 (18.8)	19 (6.7)	3.19 (1.7, 6.1)	<0.001	2.68 (1.4, 5.2)	0.004
Elevated triglycerides	94 (21)	50 (48.5)	44 (12.6)	3.47 (2.1, 5.6)	<0.001	4.33 (2.6, 7.2)	<0.001
Low HDL-Cholesterol	212 (47.3)	83 (80.6)	129 (37.0)	2.19 (1.4, 3.4)	<0.001	1.81 (1.2, 2.8)	0.009
MetS	28 (6.3)	26 (20.3)	0 (0.0)	-	-	-	-

Note: Data is presented as frequency (%). The odds ratio was calculated by logistic regression for MetS components in the group with a SPISE ≤ cut-off compared to the group with SPISE > cut-off. * Represents model adjusted with age. *p*-value < 0.05 considered as significant.

## Data Availability

The data presented in this study are available upon request from the corresponding author.

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
