# Peer review of "Sex-Specific Cut-Offs of Single Point Insulin Sensitivity Estimator (SPISE) in Predicting Metabolic Syndrome in the Arab Adolescents"

_diagnostics, 2023, doi:10.3390/diagnostics13020324_

Round 1

Reviewer 1 Report

Dear Author,

Is an interesting article that brings useful information for dealing with metabolic syndrome among adolescents.  

In my opinion  it meets the conditions for publication.

Kind regards,

Author Response

Is an interesting article that brings useful information for dealing with metabolic syndrome among adolescents. In my opinion  it meets the conditions for publication.

Response: The authors are extremely grateful to the reviewer for appreciating our investigation.

Reviewer 2 Report

GENERAL COMMENTS

The manuscript is interesting and addresses a timely topic of clinical relevance.

Some suggestions are provided to further strengthen the manuscript.

Introduction, page 2, line 66: “Regardless of, however, whether it…” is grammatically incorrect; amend.

Introduction, page 2, line 70: replace “last” by “over”.

Introduction, page 2, line 75: use abbreviatures consistently throughout the manuscript; here for example replace “insulin resistance” by “IR”.

Introduction, page 2, line 76: replace “has” by “have”.

Introduction, page 2, line 85: replace “Kg” by “kg”.

Introduction, page 2, line 89: replace “need” by “needs”.

At the end of the Introduction, page 2, line 91: formulate the specific working hypothesis – what did you expect and why.

Mat & Met, page 3, line 115: replace “kgs” by “kg”.

Results, Tables: replace “kg/m2” by “kg/m2”; do not use decimals for waist, hip systolic and diastolic BP

Results: since the authors determined both adiponectin and leptin it would be very useful and interesting to calculate the adiponectin/leptin ratio and show the data obtained given that it is reportedly a functional biomarker of adipose tissue inflammation (ref Frühbeck G, Catalán V, Rodríguez A, Ramírez B, Becerril S, Salvador J, Colina I, Gómez-Ambrosi J. Adiponectin-leptin Ratio is a Functional Biomarker of Adipose Tissue Inflammation. Nutrients. 2019 Feb 22;11(2):454. doi: 10.3390/nu11020454. PMID: 30813240; PMCID: PMC6412349).

Discussion, page 11, line 276-277: rephrase, at least as regards to visceral adiposity; insulin resistance does not really “predispose” to visceral adiposity; it is rather the other way round; visceral adiposity promotes the development of insulin resistance.

Discussion, page 11, line 313: “…and simpler tools for estimating IR and MetS” is grammatically incorrect; amend.

Discussion, page 12, line 332: “didn’t” is colloquial; amend.

Discussion, page 12, line 333-334: “…with adipose tissue derived and inflammatory markers reported here” is grammatically incorrect; amend.

Discussion, page 12, line 339: comment here the adiponectin/leptin ratio data obtained given that it is reportedly a functional biomarker of adipose tissue inflammation (ref Frühbeck G, Catalán V, Rodríguez A, Ramírez B, Becerril S, Salvador J, Colina I, Gómez-Ambrosi J. Adiponectin-leptin Ratio is a Functional Biomarker of Adipose Tissue Inflammation. Nutrients. 2019 Feb 22;11(2):454. doi: 10.3390/nu11020454. PMID: 30813240; PMCID: PMC6412349).

Discussion, page 12, line 350: given the relationship between excess adiposity and insulin resistance this should be explicitly mentioned (ref Frühbeck G, Busetto L, Dicker D, Yumuk V, Goossens GH, Hebebrand J, Halford JGC, Farpour-Lambert NJ, Blaak EE, Woodward E, Toplak H. The ABCD of Obesity: An EASO Position Statement on a Diagnostic Term with Clinical and Scientific Implications. Obes Facts. 2019;12(2):131-136).

Discussion, page 12, line 350: it should be mentioned that the probable involvement of other adipose-related factors can not be discarded. For instance, the participation on IR development of dysfunctional adiposity and its secreted adipokines with an effect on glucose metabolism and lipolytic activity should not be mentioned (refs Rodríguez A, et al. The ghrelin O-acyltransferase-ghrelin system reduces TNF-α-induced apoptosis and autophagy in human visceral adipocytes. Diabetologia. 2012 Nov;55(11):3038-50  // Frühbeck G, Gómez-Ambrosi J, Salvador J. Leptin-induced lipolysis opposes the tonic inhibition of endogenous adenosine in white adipocytes. FASEB J. 2001 Feb;15(2):333-40).

Discussion: in the limitations indicate that body fat percentage was not determined; it can be argued that the Adiponectin-leptin ratio was calculated as a functional biomarker of adipose tissue inflammation.

The manuscript needs English editing.

Author Response

The manuscript is interesting and addresses a timely topic of clinical relevance.

Some suggestions are provided to further strengthen the manuscript.

Response: The authors thank the reviewer for a thorough review which has indeed improved the quality of the manuscript

Introduction, page 2, line 66: “Regardless of, however, whether it…” is grammatically incorrect; amend.

Response: This has been amended, thanks

Introduction, page 2, line 70: replace “last” by “over”.

Response: Replaced, thanks

Introduction, page 2, line 75: use abbreviatures consistently throughout the manuscript; here for example replace “insulin resistance” by “IR”.

Response: This has been replaced throughout the manuscript

Introduction, page 2, line 76: replace “has” by “have”.

Response: Replaced, thanks

Introduction, page 2, line 85: replace “Kg” by “kg”.

Response: Replaced, thanks

Introduction, page 2, line 89: replace “need” by “needs”.

Response: Replaced, thanks

At the end of the Introduction, page 2, line 91: formulate the specific working hypothesis – what did you expect and why.

Response: The working hypothesis has been formulated and added as suggested by the reviewer

Mat & Met, page 3, line 115: replace “kgs” by “kg”.

Response: Replaced, thanks

Results, Tables: replace “kg/m2” by “kg/m2”; do not use decimals for waist, hip systolic and diastolic BP

Response: The draft has been revised as per reviewer’s suggestion.

Results: since the authors determined both adiponectin and leptin it would be very useful and interesting to calculate the adiponectin/leptin ratio and show the data obtained given that it is reportedly a functional biomarker of adipose tissue inflammation (ref Frühbeck G, Catalán V, Rodríguez A, Ramírez B, Becerril S, Salvador J, Colina I, Gómez-Ambrosi J. Adiponectin-leptin Ratio is a Functional Biomarker of Adipose Tissue Inflammation. Nutrients. 2019 Feb 22;11(2):454. doi: 10.3390/nu11020454. PMID: 30813240; PMCID: PMC6412349).

Response: The authors thank the reviewer for this suggestion… Adiponectin/leptin ratio has been calculated and inserted in the tables. A significant positive correlation of this ratio with the SPISE index was also seen and reported which is consistent with the findings reported in the reference suggested by the reviewer. This has been mentioned in the revised discussion. 

Discussion, page 11, line 276-277: rephrase, at least as regards to visceral adiposity; insulin resistance does not really “predispose” to visceral adiposity; it is rather the other way round; visceral adiposity promotes the development of insulin resistance.

Response: The authors agree with the reviewer that visceral adiposity promotes IR. This was written in the context that some researchers consider this as a to and from process. However, to avoid confusion, visceral obesity has been omitted from the sentence in the revised draft. 

Discussion, page 11, line 313: “…and simpler tools for estimating IR and MetS” is grammatically incorrect; amend.

Response: This has been amended in the revised text.

Discussion, page 12, line 332: “didn’t” is colloquial; amend.

Response: This has been amended in the revised text.

Discussion, page 12, line 333-334: “…with adipose tissue derived and inflammatory markers reported here” is grammatically incorrect; amend.

Response: Thanks for correcting, this has been amended in the revised text.

Discussion, page 12, line 339: comment here the adiponectin/leptin ratio data obtained given that it is reportedly a functional biomarker of adipose tissue inflammation (ref Frühbeck G, Catalán V, Rodríguez A, Ramírez B, Becerril S, Salvador J, Colina I, Gómez-Ambrosi J. Adiponectin-leptin Ratio is a Functional Biomarker of Adipose Tissue Inflammation. Nutrients. 2019 Feb 22;11(2):454. doi: 10.3390/nu11020454. PMID: 30813240; PMCID: PMC6412349).

Response: The comment about adiponectin/leptin ratio and its correlation with this index has been inserted with the suggested reference in this paragraph (para 5) of the revised discussion section.

Discussion, page 12, line 350: given the relationship between excess adiposity and insulin resistance this should be explicitly mentioned (ref Frühbeck G, Busetto L, Dicker D, Yumuk V, Goossens GH, Hebebrand J, Halford JGC, Farpour-Lambert NJ, Blaak EE, Woodward E, Toplak H. The ABCD of Obesity: An EASO Position Statement on a Diagnostic Term with Clinical and Scientific Implications. Obes Facts. 2019;12(2):131-136).

Response: Thanks, this point has been incorporated with the suggested reference in paragraph (para 4) of the revised discussion section.

Discussion, page 12, line 350: it should be mentioned that the probable involvement of other adipose-related factors cannot be discarded. For instance, the participation on IR development of dysfunctional adiposity and its secreted adipokines with an effect on glucose metabolism and lipolytic activity should not be mentioned (refs Rodríguez A, et al. The ghrelin O-acyltransferase-ghrelin system reduces TNF-α-induced apoptosis and autophagy in human visceral adipocytes. Diabetologia. 2012 Nov;55(11):3038-50  // Frühbeck G, Gómez-Ambrosi J, Salvador J. Leptin-induced lipolysis opposes the tonic inhibition of endogenous adenosine in white adipocytes. FASEB J. 2001 Feb;15(2):333-40).

Response: Thanks, this point has been incorporated with the suggested references at the end of paragraph (para 5) of the revised discussion section.

Discussion: in the limitations indicate that body fat percentage was not determined; it can be argued that the Adiponectin-leptin ratio was calculated as a functional biomarker of adipose tissue inflammation.

Response: Thanks, the limitation mentioned has been added in the revised draft.

The manuscript needs English editing.

Response: A thorough check with the English editing services has been done and incorporated in the revised draft.

Reviewer 3 Report

The current manuscript titled: "Sex-specific cut-offs of Single Point Insulin Sensitivity Estimator (SPISE) in Predicting Metabolic Syndrome in the Arab Adolescents" represents an important analysis of evolving field of Diabetes and Pediatrics.

The title reflects the manuscript content and helps the reader navigate the article essence.

In my opinion, these are the adjustments which should be made to increase the value of your manuscript:

1.      In Introduction chapter, please, add detailed information about studied novel biomarkers and the pathophysiological processes linking diabetes and Resistin, Leptin, Adiponectin, Vitamin D.

2.      In the exclusion criteria, please specify in detail which diseases have been excluded from endocrine, cardiovascular, gastrointestinal, and renal disorders.

3.      Line 114: please, change “mmHG” to “mmHg”.

4.      In Methods chapter, add please information about Informed Consent Statement.

5.      In the Discussion section, there is not enough comparative information with other studies.

6.       In Conclusions chapter, please do not repeat results informatio, but formulate the conclusions more clearly and precisely with practical relevance clarification.

7.       The manuscript contains some punctuation errors and typos, please revise the text (e.g., lines 27, 30 147, 149, 198, etc.).

Author Response

The title reflects the manuscript content and helps the reader navigate the article essence.

Response: The authors thank the reviewer for the appreciation.

In my opinion, these are the adjustments which should be made to increase the value of your manuscript:

  1. In Introduction chapter, please, add detailed information about studied novel biomarkers andthe pathophysiological processes linking diabetes and Resistin, Leptin, Adiponectin, Vitamin D.

Response: The authors thank the reviewer for this suggestion. This has been expanded in the second paragraph of the revised introduction section.

  1. In the exclusioncriteria, please specify in detail which diseases have been excluded from endocrine, cardiovascular, gastrointestinal, and renal disorders.

Response: The list of the common such diseases that was there in the questionnaire has been included in the exclusion criteria statement in the revised draft.

  1. Line 114: please, change “mmHG” to “mmHg”.

Response: This has been changed, thanks.

  1. In Methods chapter, add please information about Informed Consent Statement.

Response: Thanks, this has been added (section 2.1).

  1. In the Discussion section, there is not enough comparative information with other studies.

Response: The authors thank the reviewer for the suggestion. Because of the novel nature of this surrogate marker of insulin sensitivity, there are limited comparative studies on SPISE index which were referenced already and this has been acknowledged. However, alternatively, some comparative studies on associations of insulin sensitivity with adipose-derived factors have been added in the revised discussion section (especially in paragraph 4 and 5).

  1. In Conclusions chapter, please do not repeat results information, but formulate the conclusions more clearly and precisely with practical relevance clarification.

Response: The conclusion has been revised as per the suggestion of the reviewer, thanks.

  1. The manuscript contains some punctuation errors and typos, please revise the text (e.g., lines 27, 30 147, 149, 198, etc.).

Response: The manuscript has been revised with English editing services to deal with such punctuation errors and typos, thanks

Reviewer 4 Report

Metabolic syndrome is a real public health problem and needs to be managed as early as possible. Incorporating into current practice an index that predicts the occurrence of Metabolic syndrome and includes variables that are as easy to use as possible is a challenge.

Conducting such studies on populations in distinct geographical areas is essential in validating a new marker for introduction into routine practice. This new marker has the advantage of using more criteria with decreased bias risk (such as insulin in the HOMA index).

The methodology of the study is well defined, the patients included in the study  are from a specific georpgraphical area. The results are presented accurately using advanced statistical methods.

 Given that the study population consists of adolescents of both sexes the Tanner and Marshall staging of the pacients included in the study should be mentioned.

The cut-offs obtained for SPISE show population variability and support the need for further study.

The references cover essential works on the subtopic and are up-to-date.

Otherwise, it is a very well written article from a reference medical school that I hope will present us with further results in the near future.

Author Response

Metabolic syndrome is a real public health problem and needs to be managed as early as possible. Incorporating into current practice an index that predicts the occurrence of Metabolic syndrome and includes variables that are as easy to use as possible is a challenge. Conducting such studies on populations in distinct geographical areas is essential in validating a new marker for introduction into routine practice. This new marker has the advantage of using more criteria with decreased bias risk (such as insulin in the HOMA index).

Response: The authors agree with the reviewer that MetS needs to be addressed in the population actively and hence the authors chose to look for indices that predict MetS in Arab adolescents. The present index, studied in this report, was established earlier in some other population; however, it was not yet assessed in the Arab population and the authors are glad that a cut-off for this index to predict MetS was proposed here.  

The methodology of the study is well defined, the patients included in the study  are from a specific georpgraphical area. The results are presented accurately using advanced statistical methods.

Response: The authors appreciate the comments from the reviewer regarding the robust methodology and statistical analysis used in this study.

 Given that the study population consists of adolescents of both sexes the Tanner and Marshall staging of the pacients included in the study should be mentioned.

Response: The authors thank the reviewer for this suggestion. Since the study population were boys and girls of the age-range 10-17 years, pre-pubertal and pubertal development stages (five stages proposed by Marshall and Tanner) in the study population might have had some effect on the SPISE index. However; unfortunately, the study lacks the data on the factors of sexual maturity rating (SMR) that would have helped categorize the subjects into these stages. Having said that, this is indeed a good idea that could make up an interesting study in the future. For now, this has been added as a limitation in this study.     

The cut-offs obtained for SPISE show population variability and support the need for further study.

Response: The authors agree with the reviewer on this point and hence recommendations for future studies were given in the conclusion section.

The references cover essential works on the subtopic and are up-to-date.

Otherwise, it is a very well written article from a reference medical school that I hope will present us with further results in the near future.

Response: The authors thank the reviewer for the appreciation.

Round 2

Reviewer 2 Report

The authors have satisfactorily addressed the issues raised.

Reviewer 3 Report

I agree with the changes made, which significantly improve the quality of the manuscript. I recommend this article for publication.